# Position: Living Knowledge, Foundations for General Intelligence

## Abstract

We propose that general intelligence requires living knowledge – a schema-labile, autopoietic system capable of self-organization, growth, and the creation of new concepts. Grounded in language atomics and represented via *metagraphs*, we outline a possible self-organizing knowledge architecture that unifies perception, reasoning, and memory into a dynamic, emergent form of knowledge that befits general intelligence.

## 1. Introduction

Living organisms learn from reality. They use the knowledge of their environment to survive and reproduce as individuals and evolve as species. Essentially, living requires knowing. We seek to build machines that emulate such knowing: artificial general intelligence (AGI). If we wish to design an AGI system that mimics intelligent life, what aspects of living are involved and how might those be represented as technology? How can we use the idea that knowledge is living? Languages may hold the key to the answer.

### 1.1. Autopoiesis

In their seminal work, Humberto Maturana and Francisco Varela argue that living systems are autonomous, self-creating, and self-maintaining machines (*autopoietic*), and that cognition is an inherent biological phenomenon of all living systems (Maturana &Varela, 1972). Their theory proposes a radical shift in viewing life, not as an object of external observation, but as a self-contained, self-referential process, where cognition is the very nature of a living system's interaction with its environment. Further advancing this concept, their later work (Maturana &Varela, 1984) explores how knowledge and understanding are biologically rooted. It argues we don't simply represent a pre-existing world, instead, our experience of the world is an active, ongoing creation that arises from our biological and social history. Their theory concludes language is employed by humans not just as a tool for communication, but also as a social-structural coupling that creates our shared reality: we can only understand what we have learned through this shared social experience.

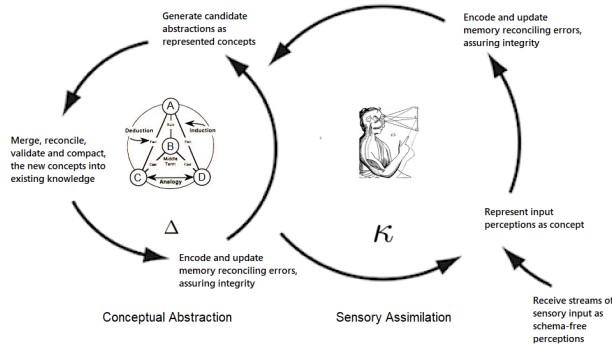

*Figure 1.* The conceptual stance on the cognitive process in intelligent agents.

In other words, a general, living form of knowledge imbues the cognitive design of all living things: "living knowledge." Knowledge grows over time from new inputs, it is self-referential: introspecting its own contents and operations, and it evolves by reorganizing as new relationships are discovered. Living knowledge is autopoietic: capable of generating new structures from within.

### 1.2. General Intelligence

*General intelligence* remains elusive in machine expression. Machine Learning (ML), Deep learning (DL), reinforcement learning (RL), and large language models (LLM) exhibit strong performance in domains that benefit from extensive training data, such as general knowledge, language comprehension, and mathematical reasoning. Despite their impressive recent progress, they remain limited in achieving task generality, knowledge transfer, robustness, and open-world reasoning. Several studies reveal that these shortcomings primarily stem from the absence of long-term memory and continual learning mechanisms, resulting in a form of functional amnesia that forces models to repeatedly reconstruct context during each interaction (Hendrycks et al., 2025).

Our position contends that **general intelligence must entail living knowledge because generality of intelligence emerges from autopoiesis of knowledge**.

### 1.3. Document Structure

This paper explores the mechanisms underpinning living knowledge. Using natural languages as a proven case of

living knowledge, it investigates how languages appear to effortlessly express open-ended, evolving concepts. Thereafter, recognizing that current digital knowledge systems (knowledge graphs, databases, symbolic logic & deep neural network) remain brittle and rigidly schema-bound, the study seeks to unravel the mechanics necessary to support language-like fluidity in the representation of knowledge for the next generation of artificial intelligence (AI).

We start out our discussion studying two opposing views of the mental machinery, juxtaposed against each other in Fig 1. The right-hand side of the figure is René Descartes's mechanical view of cognition, wherein the mind functions as a reactive process of sensory assimilation. This process-centric view (denoted by the Greek letter $\kappa$) broadly influences modern agentic design in AI. On the left side is *Aristotle's Paradigm* (in *Prior Analytics* 2.24, also further detailed in Figure 2), that emphasizes a deliberative process of reasoning (denoted by $\Delta$) that broadly occupies philosophical and mathematical discourse on logic. In this paper, we show that $\Delta$ and $\kappa$ are complementary processes and together they form the basis for living knowledge. We begin with definitions and introductory concepts. Then we present the main position: language forms the basic model for living knowledge. As counterpoint, we present alternative views and discuss their shortfall. Finally, a discussion section summarizes the research, details open questions about technical implementation and other challenges before drawing final conclusions.

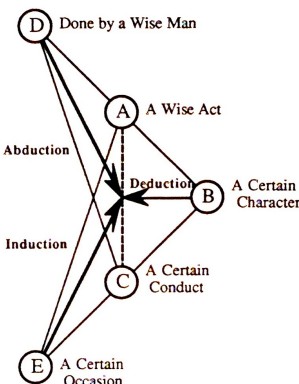

*Figure 2. Aristotle's Paradigms* conceptualized as interrelated transitional processes by Charles S. Pierce (Peirce, 1865).

## 2. Definitions

Within this paper, we use the following terms to have explicit meanings, narrower than their conventional usage:

- *planning process*, a model-based procedure that constructs and evaluates action sequences under predicted dynamics to reach a goal under constraints.
- *deliberative process*: a controlled inference-and-search routine that explicitly represents alternatives and selects beliefs or actions by optimizing over structured intermediate reasoning.
- *language*: a compositional system of discrete primitives and operators governed by constraints that enables the systematic construction and interpretation of meanings.
- *concept*: an atomic, abstract, generalizable unit of meaning that represents a particular instance entity or a general class of entities or relations between entities across contexts.
- *conception*: a collection of related concepts that share semantic proximity and theoretic representation of a main primary concept - the concrete assumptions, features, and inferential roles that makes it usable.
- *schema*: a formal specification of allowable and valid structures describing data; whereas a schema constrains, a structure realizes.

## 3. Preliminaries

### 3.1. What is General Intelligence?

General intelligence refers to the capacity of an agent – biological or artificial – to learn, reason, and solve problems, particularly problems in unfamiliar contexts. AGI aims to replicate this broad competence, distinguishing itself from the current, narrow form of AI that is confined to specialized tasks. The distinguishing features of AGI fall into categories of (i) adaptability, (ii) abstraction and generalization, (iii) continual learning, (iv) the invention of new concepts, and (v) the ability to project and transfer knowledge across domains. These five classes of capabilities depend on effective knowledge management (KM) and knowledge representation (KR) as the foundation for reasoning and learning. Whereas KM encompasses processes such as: capturing and creating knowledge, organizing and storing knowledge for efficient retrieval, and sharing and disseminating knowledge across systems, its overlapping counterpart KR involves encoding knowledge in formal, machine-interpretable formats that are surrogates for real-world understanding. This paper focuses on the latter: knowledge representation as a critical foundation for general intelligence.

### 3.2. Knowledge Representation

The creation of comprehensive taxonomies or classifications of things occupied the ancients. Early attempts emphasized classification and categorization schemes to organize knowledge about the world. In Western thought, knowledge representation is rooted in Greek philosophy, with our modern stance tracing its expositional origins to 300 BCE in Aristotle's *Metaphysics*. Aristotle's *Organon* includes the treatise *Categories*, the earliest attempt to define a comprehensive upper ontology (Fig. 3). It employed the device of *binomial*

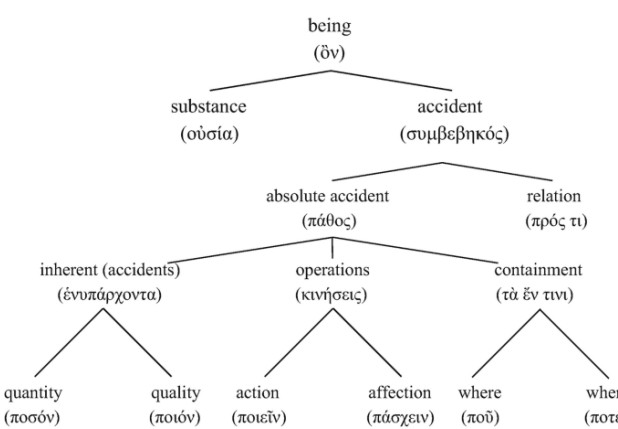

*Figure 3.* Aristotle's *Categories* (Categories 1b25), an early form of an upper ontology (Raspa, 2020).

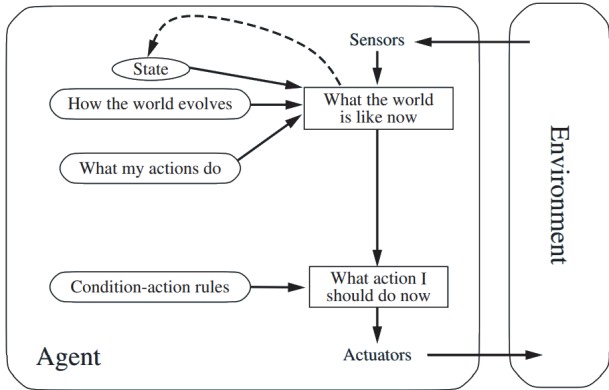

*Figure 4.* A model-based reflex agent, a type of AI agent that operates in partially observable environments by maintaining and referring to internal world models. (Russell &Norvig, 2010).

*nomenclature* – a structure of genus and species – to support increasingly fine-grained classification. This approach led to taxonomies with hierarchical tree structures, the preferred model for knowledge representation from the Middle Ages until recently.

With the advent of expert systems (e.g., OpenCYC, NextKB) and web-scale efforts (e.g., DBPedia, Google Knowledge Graph) able to develop comprehensive ontologies, taxonomies gave way to richer graph-based representations that better captured reality (Kejriwal et al., 2021). The philosophical challenge of defining natural "kinds" of things, and handling fuzzy category boundaries, remains unresolved (Wittgenstein, 1953; Quine, 1953). Recent advances in connectionist or sub-symbolic systems, such as deep neural networks (DNN), avoid many issues tied to rigid ontologies in symbolic systems, but they do so at the cost of precision. Blending the two approaches is practical for specific use cases, but without clear boundaries, it becomes challenging to determine how best to combine the flexibility of neural networks with the exactness of logical semantics for effective knowledge representation. A recent resurgence in neuro-symbolic AI takes this approach, seeking to combine the flexibility of connectionist approaches with the precision of symbolic ones, but predictably faces challenges such as creating unified representations, ensuring interpretability, and scaling efficiently. Neuro-symbolic AI also struggles with handling uncertainty, maintaining knowledge bases, and achieving seamless integration between continuous learning and discrete logic (Zhang &Sheng, 2024).

### 3.3. Machine Reasoning

Today, there are three main approaches to the study of mental objects: (1) classical *modal logic and possible worlds* framework (Fig. 4), rooted in philosophy, models knowledge in terms of what is true across possible scenarios (Russell &Norvig, 2010); (2) the *first-order fluents* approach

treats mental states as time-varying properties within a logical system, enabling dynamic reasoning. Although it applies possible-world semantics, it provides operationally grounded reasoning frameworks for applied AI; and (3) *syntactic theory* that represents mental content as symbol strings, wherein the mental content is a manipulable knowledge structure. Each approach differs in semantic rigor, dynamic modeling, or computational structure, and each informs different areas of AI and cognitive science, from commonsense reasoning to temporal goal specification in learning systems.

### 3.4. Language Theory

**Language is the most accessible manifestation of human cognition, serving as a primary observable for general intelligence research**. Unlike fixed symbolic schemas, language exhibits properties that reflect the open-ended nature of human knowledge: (i) it is *malleable*, allowing meanings to drift and adapt across contexts; (ii) it is *synthetic*, enabling the creation of novel concepts and abstractions; and inherently *ambiguous*, with polysemy and adversarial interpretations that befuddle rigid formal systems, and (iii) language is boundlessly *emergent* and evolving - constantly synthesizing new schemas, genres, and morphologies over time. Despite these differences, written languages are exemplary candidates for the practical study of functioning living knowledge and therefore knowledge representation for AGI must incorporate mechanisms that mirror the flexibility and creativity of languages.

## 4. Position: Knowledge as Autopoiesis

Knowledge in a general intelligence system must be autopoietic, capable of growing and reorganizing its structure from empirical evidence about the world, instead of relying on a fixed, predefined schema. Traditional systems impose rigid concepts in advance, limiting adaptability and long-term

generalization, and resulting in systems that are inherently incomplete and limited. In contrast, we claim that **generality of knowledge requires the ability for schemas to evolve as a consequence of both use and learning**. *Living knowledge* emerges from the accumulation and reorganization of knowledge in response to ongoing interaction with the environment. This mirrors the property of autopoiesis inherent in biological systems (Maturana &Varela, 1972), wherein a living system actively maintains and extends its own organization from internal components, without relying on externally imposed structure.

For analogous digital knowledge systems, this means continuous integration of new information, while (a) identifying anomalies, (b) reorganizing memory to assure consistency, and (c) promoting or suppressing memory based on internal utility. Consequently, an AGI system aspiring to a level of general intelligence comparable to living things, must satisfy the following six key properties: (1) be open and adaptive to novel inputs, (2) grow dynamically over time, (3) self-organize its world model, (4) self-correct based on empirical feedback and predictive error, (5) employ evolutionary strategies of selection based on epistemic utility to evolve conceptual schemas, and (6) decay knowledge: time-based degradation for "garbage-collection" of concepts with diminishing utility. To realize these properties, the system must include a universal language.

## 5. Atomics of Language as Knowledge Primitives

Languages allow us to function in our world – to describe the world around us, to direct our interactions with it, and to share knowledge with others. Hence we contend that three classes of atomics are sufficient to express all truth-apt statements – **the full range of knowledge required for general intelligence is constituted of: (1) things as they are, (2) things in relation to other things, and (3) ways of employing or controlling things.** The first category of statements provide an intrinsic descriptions of an entity's properties. The second category for statements encodes extrinsic structure; they introduce relations, classifications, collections and composite structures, as well as spatial and temporal organization. The last category concerns their use and control of things; statements about employment: deontic claims about what must or may be done, modal and probabilistic claims about what might or must be the case, counterfactuals about what would happen under alternative conditions, and dynamic speech acts that change norms or states by being stated. Together, they furnish a minimal but expressive set of atomics from which almost all rich knowledge structures can be composed.

Our scheme develops the three categories of statements into the following seven *atomic* symbols representing various

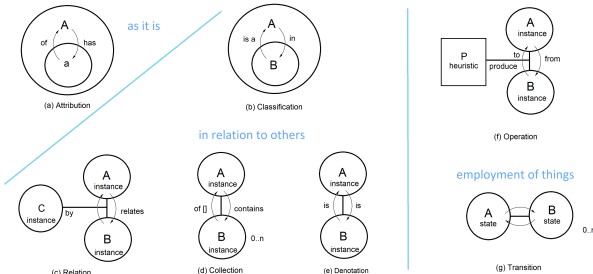

*Figure 5.* Three classes of structures manifest as seven primitives, the *linguae primae* that underlie how knowledge is represented in language: a piece of knowledge *as it is*, then *as it relates to other* knowledge, and how it is *employed*. Relating includes classification (hierarchies, schemas), relationships (links, and how links are modified by metadata), and how instances of knowledge can be part of non-hierarchical collections.

forms of language structures.

- *Attribution* (Property) – Assigning a property to an entity (e.g., "Socrates is wise").
- *Classification* (Type) – Placing an entity under a category (e.g., "Socrates is a man").
- *Relation* (Association) – Stating an association that does not reduce to attribution or causation (e.g., "Plato was taught by Socrates").
- *Collection* (Aggregation) – Placing an entity under an instance of a collection (e.g., "Socrates is a Greek").
- *Denotation* (Identity) - Identifying two instances referring to the same underlying entity (e.g., "Leonidas is King").
- *Operation* (Transformation) – Defines an action as a production rule, employing a process that transforms one entity into another. (e.g., "Wood into fire").
- *Transition* (Causation) – Establishing a conditional dependence between events (e.g., "Fire causes smoke").

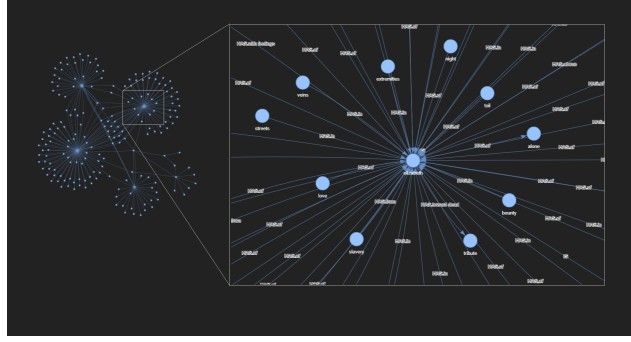

*Figure 6.* The narrative structure of the main characters in Mary Shelly's Frankenstein exhibit natural composite floret-like structures when transformed to visual metagraphs using *linguae primae*.

### 5.1. Supporting Language Constructs

The preceding core atomics are augmented by three classes of connectors and contextual assistance that provide a "glue"

for joining the language primitives: (1) *logical operators* function as a connective tissue for knowledge, augmenting its substantive content: they bind together intrinsically and extrinsically focused statements, regulate how they compose, and constrain what follows from what, without themselves introducing new objects or properties into the world; (2) *context and indexicals* ("I", "here", "now"); and (3) *non-declaratives* (questions, commands, exclamations) that are necessary for situating and enriching context, but remain auxiliary to the underlying representational core, without changing the basic inventory of things, relations, and uses that the knowledge system must encode.

### 5.2. Reasoning with Atomics

Concept invention and schema emergence require advanced analytical reasoning to synthesize abstract conception. Topological operations over sets of vertices are elevated to graph topology, amenable to inductive logic, set semantics, and De Morgan's relative product. Aristotle's work, particularly *Prior Analytics*, provides a formal system of deductive logic that can be applied to metagraphs. Whereas graph mining techniques employing traversal methods, isolating patterns and power laws allow the detection of nodes of consequence, incorporating inductive and abductive logic provides a wide range of tools to deduce abstract concepts of logic (Fig. 2). Pierce's collected essays on Pragmaticism entitled *Illustrations of the Logic of Science* (Peirce, 1865) offer tools to employ Aristotelian syllogism and Boolean logic to a sets of conception for abstract concept distillation. A simple example in Appendix D demonstrates this device. Note that such devices can themselves be described as heuristics represented as knowledge within as an *Operation* primitive (Fig. 5) in *linguae primae*, thus codifying self-emergent behavior within the language itself.

### 5.3. Sufficiency of the Atomics

It is hard to discern the the veracity of our claim to the sufficiency of the seven atomics without further study. However, early experiments encoding classic literature reveal rich language structures in their text (Fig. 6), leaving open the intriguing possibility of harvesting conception using nested learning approaches (Kusupati et al., 2024). As such, descriptive statements in a language-like *linguae primae* supply the object-level inventory of properties and kinds that represent the physical world – relational statements encode structure, dynamics, and dependency, and use/control statements capture norms, goals, uncertainty, and counterfactuals. Collectively, description and relational statements span what an agent can know, how the world is organized, and what can or should be done in it. This three-way decomposition of language mirrors the categories of Kant conceptualism (A70/B95–A93/B109) (Kant, 1998): "substance and quality" (thing as it is), "relation" (between things),

and "modality" (necessity, possibility, permission and the like), thus offering a computational counterpart to his claim that any intelligible experience must be organized along these axes. Supporting constructs: logical operators and context/indexical machinery, are best treated as scaffolding wrapped around these atomics of content. A foundation for general intelligence may, by this, implement rich logic, context sensitivity, and speech-act dynamics while keeping its core representational vocabulary minimal, compositional, and aligned with these three fundamental forms.

## 6. Alternative Views

This section reviews major approaches to knowledge representation and reasoning. We contrast them along three points of view relevant to our position: (a) whether schemas are predefined or emergent, (b) how knowledge changes over time, and (c) whether architectures supports self-organization.

### 6.1. Symbolic Schema-first Knowledge Representations

Conventional approaches rely on structured, predefined schemas that organize information into hierarchical and relational structures, such as semantic networks (Quillian, 1967), frames (Minsky, 1974), and entity–relationship models (Chen, 1976). These methods support automated reasoning through formal systems including first-order logic (McCarthy et al., 1960), description logic (Baader, 2003), and probabilistic reasoning with Markov logic networks (Richardson &Domingos, 2006) and Bayesian networks (Pearl, 2014). The advantage of this approach is strong semantic guarantees. However, schema-first systems are difficult to scale. They require heavy manual engineering and lack flexibility for dynamic, long-term concept evolution.

### 6.2. Neural and Neuro-symbolic Representations

Neural or connectionist approaches such as ML and DL methods eschew explicit symbolic schemas entirely and rely on statistical representations. Examples include Word2Vec (Mikolov et al., 2013), graph neural networks (Kipf, 2016), and Transformer architectures (Vaswani et al., 2017) excel at adaptability and pattern recognition but sacrifice interpretability and precise symbolic structure. Updating knowledge often requires retraining, which is costly at scale. Neuro-symbolic AI attempts to combine distributed learning with symbolic reasoning (Besold et al., 2021). Examples include memory-augmented networks (Graves et al., 2014) and neural theorem proving (Rocktäschel &Riedel, 2017). These systems partly meet generality requirements by enabling compositional and statistical reasoning, but they still depend on fixed symbolic schemas and lack autonomous schema emergence or self-organization. Their focus remains task-specific rather than fostering a dynamic knowledge pro-

cess.

### 6.3. Architectures with Structured Generative Models

Cognitive architectures such as SOAR (Laird et al., 1987), ACT-R (Anderson et al., 1997), and CLARION (Sun, 2006) model human cognition using symbolic production rules, buffers, and modular memory. They align with alternative dual-process theories, distinguishing fast, reactive reasoning from slow, deliberative reasoning (Kahneman, 2011). These approaches provide interpretability and stable reasoning structures. However, their fixed rule systems and predetermined memory modules do not support meta-level reorganization of schemas. They are designed for consistency, not conceptual self-rewrite.

Predictive processing and active inference models treat intelligence as hierarchical prediction error minimization (Friston, 2005; 2010). These architectures use dynamic generative models that adapt over time and support planning, imagining counterfactuals, and compressing experience. World-model approaches in reinforcement learning follow the same philosophy (Zhang et al., 2021). Although these approaches allow flexible inference, they still operate over fixed representational hierarchies. The schemas may adapt, but they do not fundamentally reorganize themselves or generate new instances of conceptual language primitives.

### 6.4. Deep Learning Knowledge Management

Neural networks based learning systems, including LLM and recurrent neural networks (RNN), manage knowledge primarily through their internal parameters or external vector databases. Developers often equip LLM agents with tools-scripts, APIs, or functions and define specific personas to improve task execution. Techniques such as fine-tuning, long-context attention, embeddings, RAG pipelines, and knowledge-graph retrieval improve grounding and reduce hallucination. Agentic AI frameworks extend this by orchestrating multiple agents and adopting strategies such as chain-of-thought (Wei et al., 2022), tree-of-thoughts (Yao et al., 2023), self-critique (Madaan et al., 2023), ReAct (Yao et al., 2022), and reinforcement-based adjustments (Shinn et al., 2023). These methods aim to approximate human-like reasoning on top of advanced neural networks for DL including LLM, CNN, GAN and RNN (Kim et al., 2025). Despite progress, knowledge representation remains shallow: most is stored as embeddings without symbolic structure. Knowledge graphs, when used, act as external add-ons rather than integrated, self-organizing prototypes. Temporal knowledge is problematic—models quickly become outdated and need costly retraining. LLM-centric systems lack a stable mechanism for maintaining and evolving knowledge.

### 6.5. Deductive Summary

Across these five approaches – symbolic, neural, cognitive, predictive, and LLM-based – a common theme emerges: existing systems depend either on predefined ontologies, fixed schemas, or static representational substrates. These constraints make them scalable only when knowledge remains relatively stable and domain-specific.

In our view, general intelligence requires long-term, large-scale knowledge management. As knowledge grows, the cost of maintaining a large and rigid schema grows with it. In contrast, as we propose in Section 3 and 4, the larger the knowledge system, the smaller and more flexible the representational building blocks must be. Without schema-emergent mechanisms, knowledge representation becomes brittle and unmanageable at scale.

## 7. Technology for Living Knowledge

Here we explore the challenges of implementing a practical system of living knowledge employing *atomics*, opening research questions about (a) the data structures required to represent it, and (b) the systems implementation required to assure it is continuously learning and self-organizing: autopoietic.

### 7.1. Metagraph Representation for Living Knowledge

Traditional graphs constrain expression with fixed edge types and rigid ontologies. Hypergraphs improve grouping but they struggle to encode meta-statements (statements about statements) or model adaptive schemas. Classical databases, though efficient, are schema-first and struggle with conceptual fluidity or novelty. To overcome the limitations of those three approaches, we advocate a *metagraph*, a graph variant formally defined as:

$$G_M = (V, E, \phi)$$

where:

- $V$ is a finite set of vertices,
- $E \subseteq V \times V$ is a set of directed or undirected edges,
- $\phi : E \to V$ is a metadata mapping that associates each edge $e \in E$ to exactly one vertex $v_e \in V$, acting as a *metadata anchor*.

In a *metagraph*, each edge $e = (u, v) \in E$ connects vertices $u$ and $v$, but its metadata is not embedded within the edge itself, but attached to a third vertex $\phi(e) \in V$. Metagraphs treat vertices as not only entities or concepts, but also as statements, allowing edges to represent intrinsic, extrinsic, and functional relations intrinsically. Additionally, annotations and metadata are elevated to first-class graph citizens, enabling emergent semantics and self-refining knowledge structures. Hence, knowledge operations such as infer-

ence, compression, or adversarial reasoning become graph rewrites and traversals. *Metagraphs* are also conceived to serve not merely as data models but as dynamic infrastructures for semantic evolution - forming the foundation of a self-organizing knowledge architecture (SOKA), where knowledge is not static but dynamic.

### 7.2. Self-Organizing Knowledge Architecture

Given that we have established the basis for language atomics and a data structure to represent it efficiently, an important question still remains unanswered - how would a SOKA system work? The diagram in Fig. 7 shows an architecture sketch of a self-organizing knowledge architecture for an agent in continuous interaction with an open environment.

*Reactive Process*, ($\kappa$): The system perceives the real environment through sensors - "perceive & encode", forming sensory representations, of experienced phenomena, that flow into memory holding a learned world model. An agent, through planning and decision-making act on this world model to choose actions, employs actuators to execute the plan and change the environment, thus closing the observe–act loop. This is a fast, real-time, sensory-facing pipeline forming the primary reactive reasoning loop $\kappa$, that streams observations into the memory and uses its cumulative aggregate knowledge to guide the responsive behavior of a reflex agent (Fig. 4).

*Deliberative Process* ($\Delta$): Within the SOKA architecture, the reactive reasoning loop is complemented by a concurrently running second loop $\Delta$ of deliberative reasoning to construct abstract knowledge. Inside the memory of the agent, knowledge is stored as a structured *metagraph* that can be *sliced* into local subgraphs (Fig. 7 numbers $1 \rightarrow 2$) representing candidate concepts or situations. Within the deliberative process loop, these slices are processed by a set of operations on language atomics (3), *fitted* against current anomalies or predictive needs, and *bound* and *composed* into richer structures (4). Some compositions may reveal conflicts and inconsistencies (5), while stable, epistemically consistent compositions, yet empirically ungrounded, concepts are sent into an "incubation" phase (6), where they are tested over time, validated by empirical evidence for usefulness, coherence, and predictive power. Successful candidates, validated empirically, emerge as *legitimized* and are integrated back into memory, while invalidated candidates decay over time or are discarded. This implements two steps, *accrete* and *generate*: integration, consolidation, and hypothesis formation driven by anomaly and epistemic utility.

Together, the processes of the architecture behaves like a Darwinian system over concepts that seeks to acquire and construct knowledge: slice–fit–legitimize plays the role of mutation/crossover–evaluation–selection on composite

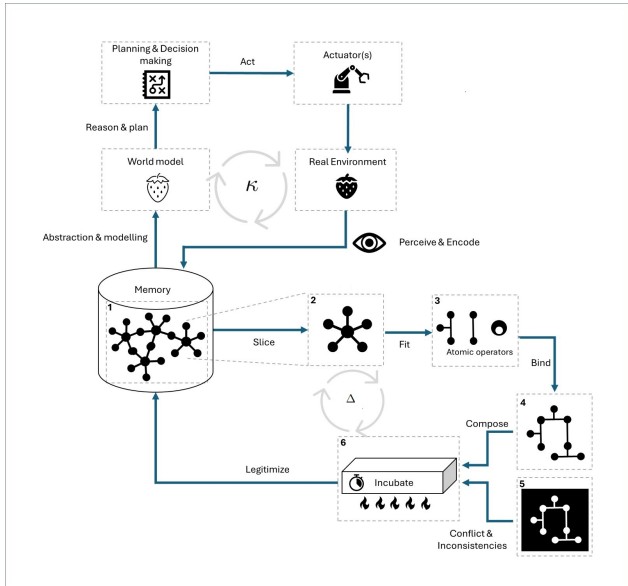

*Figure 7.* Architecture sketch of a self-organizing knowledge system.

knowledge structures. The system mirrors the psychoanalytical model outlined in Fig. 1, where likewise an Aristotelean deliberative reasoning ($\Delta$) process corresponds to this slower, deliberative layer of conceptual abstraction that lives over the same metagraph enriched by a Cartesian reactive reasoning ($\kappa$), which corresponds to active sensory assimilation of experienced phenomena. As both systems share a vocabulary of atomics and storage memory, the agent can hence continually update its world model, invent new concepts and schemas, and yet maintain global consistency through local operations. The result is a continually adapting, non-stationary knowledge process with a built-in notion of a concept life cycle and "survival of the fittest" concept abstractions. The two processes are compared and contrasted in more detail, in Appendix B.

## 8. Discussion

The discussions in the preceding sections introduce no significant novel concepts to the discourse on the mechanics of intelligence. Instead, we invite the reader to explore a fresh reinterpretation of Aristotelian and Kantian epistemology, developing a sketch for a workable foundation in modern intelligence architecture. Seeking a universal solution, we attempt a reconciliation between advances in mathematical logic (Boole, 1854) (Frege, 1879), (Peirce, 1865), psychoanalytic and cognitive accounts of the mind (Freud, 1923), (Kahneman, 2011), and neurolinguistic perspectives (Chomsky, 1957), (Pinker, 1999), arguing that language may hold a crucial key to understanding how machines eventually achieve general intelligence.

Living knowledge, as developed in this paper, is an autopoi-

etic, schema-labile system in which concepts, structures, and even ontologies emerge and reorganize subject to experience and use. Grounded in language atomics and implemented via metagraphs, SOKA (Self-Organizing Knowledge Architecture) unifies perception, reasoning, and memory into a single, evolving world model rather than a fixed database of facts. This framework emerges out of a reconciliation of Descartes' mechanical view of cognition with Aristotle's paradigm of reasoning, and builds on their principles. Closing the loop, we once again refer to Fig. 1. The fast, sensory-facing $\kappa$ loop realizes a Cartesian picture of the mind as a reactive mechanism that assimilates inputs and guides behavior in real time. The slower $\Delta$ loop realizes an Aristotelian picture, operating over the same metagraph to construct, test, and refine higher-order concepts and inferences. Their interaction – reactive grounding plus deliberative reorganization, turns knowledge into a living process: speculative, sometimes "hallucinatory," but continually corrected by empirical feedback.

Adopting the a SOKA introduces several concrete requirements for future AGI architectures:

- **A unified, schema-labile, yet structured substrate**. Knowledge –represented as atomics, statements, and metadata – coexists in a single metagraph-like composite structure enabling symbolic precision, neural-like flexibility, and meta-cognitive reflection.
- **Dual-loop cognition over a shared memory**. Architectures should (a) explicitly separate and coordinate a fast reactive loop $\kappa$ and a slower deliberative loop $\Delta$, both operating over the same shared living knowledge; (b) realize, in computing terms, the reconciliation of reactive reasoning for fast response and deliberative reasoning for conceptual understanding; and (c) employ a shared representational vocabulary for linguistic and non-linguistic experience to support both internal world modeling and external communication.
- **Eventual consistency through continuous self-correction and reinterpretation**. Knowledge must be revisable at all levels, down to its most primitive conceptual constructs. Filters for internal consistency, mechanisms for belief revision, and processes for reinterpreting preexisting data under new concepts, are primary design features. With such devices, the system prioritizes and constrains the combinatorial explosion of possible conceptual constructs during reorganizations.

## 9. Conclusion

This paper has argued for an intelligence architecture grounded in the compositional mechanics of language: *primitives* that anchor reference, *operators* that transform and bind representations, and higher-order *structures* that support abstraction, generalization, and control. Rather than introducing an entirely new theory of mind, our position reframes language and its epistemic commitments into topological constructs, a shape of knowledge – about the categories, rules, and conditions for intelligibility. This approach reconciles the formal discipline of logic with the ambiguous constraints of cognition and learning, allowing topological analysis of linguistic structure to form the primary interface where reactive agency and deliberative reasoning can be unified into scalable machine intelligence. This transforms linguistic analysis of natural language into a mathematical problem of topology analysis in latent space.

Several open questions remain and will determine whether our position can be operationalized: What is the minimal schema, the smallest set of primitives, operators, and structural constraints from which robust generalization and planning can emerge? Can such a system begin from an effectively empty state i.e. a computational *tabula rasa*, or must it be seeded with structures that resemble grammar, logic or agency? How should we formalize meta-operators or operators that construct new operators, while ensuring semantic stability? Can generative AI play a central role in synthesizing candidate primitives and structures, thus participate directly in knowledge-structure generation? Other open questions are performance considerations: Will a Darwinian selection regime - prioritizing recency over learned knowledge, work as a proxy for environmental relevance? How must the lifecycle for knowledge - its archival and obsolescence, be treated? Answering these questions clarify whether language is merely a convenient medium for intelligence or a fundamental constitutive mechanism for building general intelligence.

Concluding this report, we argue that progress toward AGI will require technologies capable of supporting an autopoietic structure for representing dynamic, self-maintaining knowledge: living knowledge. DL systems emulate the capacity for conceptual evolution, with limits on transparency and explainability, and the alternative prevailing modeling practices tend to freeze, externalize, and consequently constrain conceptual evolution. To move beyond these limitations, we propose *linguae primae* – atomics of language as knowledge primitives, and a metagraph-based representational substrate, forming the basis of a self-organizing knowledge architecture that can accommodate conceptual autopoiesis – the dynamics of concept evolution and schema emergence. This work should be read not as a complete or deployable system, but as an architectural and philosophical position: a set of commitments about how knowledge must be structured, transformed, and sustained if artificial systems are to approach general intelligence.

## Acknowledgments

**Do not** include acknowledgments in the initial version of the paper submitted for blind review.

## Impact Statement

This paper presents work whose goal is to advance the design of computer architectures for machine learning. There are many potential societal consequences of our work, in particular, how knowledge is broadly represented, aggregated, and interpreted. However, given the foundational nature of the subject matter, none of the consequences may be specifically highlighted here.

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

# A. Formal Sketch of a Knowledge Architecture

Conceptually, the function of Self-Organizing Knowledge Architecture (SOKA) is to produce knowledge from inputs, given a representation operator $\mathcal{R}$ and a knowledge extraction operator $\pi$, as follows.

$$Input \xrightarrow{\mathcal{O}} Memory \xrightarrow{\pi} Knowledge$$

This concept involves following definitions.

**Formal Definition:** Let $\mathcal{S}$ be a system that interacts with an open, partially observable environment $\mathcal{E}$ over time $t \in \mathbb{R} \geq 0$. Then, a SOKA is defined as a 9-tuple (nine components):

$$\mathcal{S} = \langle I, M, \mathcal{R}, \mathcal{L}, U, C, F, G, \pi \rangle$$

where:

- $I$ is an Input Stream, defined as:

$$I : \mathcal{O}_{t \geq 0} \mapsto \mathcal{X} \tag{1}$$

  a potentially unbounded, schema-labile stream of raw observations $\mathcal{O}(t)$ from the environment $E$, at time $t$ and mapped to a discrete representation $X^*$, in representation space $\mathcal{X}$. Streams may include elements of sensory signals, symbolic input, and/or structured/unstructured data.

- $\mathcal{L}$ is *linguae primae*, the language of primitives used to describe elements of the the representation space $\mathcal{X}$. These elements are denoted as a set of propositions, composed as triples - subject $s$, verbs $v$, and objects $o$, such that:

$$\mathcal{X} : \{e_i = (s, v, o) \mid e_i \mapsto \mathcal{L}, i \in \mathbb{N}\} \tag{2}$$

- $M$ is a Memory Structure, such that:

$$M(t) = \{m_i \in \mathcal{K} \mid i \in \mathbb{N}, m_i = f_i(I_{\leq t})\} \tag{3}$$

  is a dynamic and extensible memory containing knowledge elements $m_i \in \mathcal{K}$, where $\mathcal{K}$ is the knowledge representation space (e.g., vectors, graphs, symbolic expressions). Irrespective of the underlying storage structure it is constructed from, $M$ must support retention, modification, composition, and compression.

- $\mathcal{R}$ is the Representation Operator, such that:

$$\mathcal{R} : X^* \times M \mapsto \mathcal{K} \tag{4}$$

  is a representational mechanism that maps sequences of observed representations $X^*$ projected against current aggregated memory to new knowledge representations in the knowledge space $\mathcal{K}$. $\mathcal{R}$ relies on the capabilities of the *linguae primae* $\mathcal{L}$ to enable composability, abstraction, and structural generalization without reliance on a fixed schema.

- $U$ is the Update Dynamics, such that:

$$U : M \times K \mapsto M' \tag{5}$$

  governs the incorporation of new knowledge into the system's memory. $U$ satisfies the following non-stationary growth condition on memory $M$, ensuring that knowledge is strictly accumulating or restructuring during the lifecycle of the system:

$$\exists t_1 < t_2 : |M(t_1)| < |M(t_2)| \wedge M(t_2) \nsubseteq M(t1)$$

- $C$ is a Anomaly Check Function such that:

$$C : M \times \mathcal{X} \mapsto \mathbb{R} \geq 0 \tag{6}$$

  is an intrinsic utility function that estimates the epistemic value of new inputs or knowledge states. $C$ is a heuristic function that drives attention, novelty-seeking, and active sampling behavior on the aggregate knowledge of the environment, by assigning high value to input that leads to prediction error, uncertainty reduction, or hypothesis expansion. It serves a function analogous to dopamine surges in response to new experiences within mammalian brains.

- $\pi$ is a Knowledge Extraction Operator defined as:

$$\pi : K \mapsto K' \tag{7}$$

  as the threshold of anomalies detected by $C$ is crossed, the extraction operator $\pi$ generates high-order knowledge structures with locality, acting on a subpart of $\mathcal{K}$, while preserving its soundness - semantic integrity within $\mathcal{K}$. In essence growing the knowledge structure through abstract analytic reasoning - abductions, deductions and inductions.

- $F$ is the self-correcting Consistency Filter, such that:

$$F : M \mapsto M', K \mapsto K' \tag{8}$$

  whereas the extractor $\pi$ expands the knowledge through abstract reasoning, it may in the process synthesize conflicting inferences. The consistency filter detects and attempts to resolves those internal contradictions, redundancies, or incoherencies in memory. However, it can only enforce a weak form of epistemic consistency. But it may support processes such as belief revision, memory compression, concept obsolescence and structural reorganization.

- $G$ is a Knowledge Structure Synthesis Operator, such that:

$$G : \mathcal{S} \times M \times \mathcal{K} \mapsto \mathcal{X} \tag{9}$$

  providing a capacity for concept invention and schema emergence, it synthesizes candidate representations for further self-organization and evolution from within the system $\mathcal{S}$. It does this through generative language devices to produce novel fabricated representations $X^* = \hat{X}$, that are neither a product of experience nor a product of analysis, but are nevertheless plausible.

Using the formal definition above, the pseudo-code for the knowledge generation component of the reactive reasoning $\kappa$, that processes and encodes information from the environment into memory, of the SOKA architecture diagram in Fig.7, is as follows:

---

**Algorithm 1** REACT()

---

1: **Initialize:** ;
2: $M_0 \leftarrow \{\emptyset\}$
3: $K_0 \leftarrow R_0 \leftarrow \{\mathcal{I}\}$
4: **while** true **do**
5: $\quad X^* \leftarrow I(t)$ {perceive}
6: $\quad K' \leftarrow \mathcal{R}(X^*, M_t)$ {encode}
7: $\quad M' \leftarrow U(M_t, K')$ {update memory}
8: $\quad \hat{M}, \hat{K} \leftarrow \text{ACCRETE}(X^*, M', K_{t+1}, \alpha_{\text{anomaly}})$
9: $\quad (M_{t+1}, K_{t+1}) \leftarrow F(\hat{M}, \hat{K})$ {self-correct}
10: **end while**

---

Likewise, the pseudo-code for the knowledge generation component of the deliberative reasoning $\Delta$, that synthesizes abstract meta knowledge and codifies it into memory, in Fig.7, is as follows:

---

**Algorithm 2** DELIBERATE()

---

1: **while** true **do**
2: $\quad X^* \leftarrow G(S, M_t, K_t)$ {synthesize candidates}
3: $\quad \hat{M}, \hat{K} \leftarrow \text{ACCRETE}(X^*, M_t, K_t, \alpha_{\text{anomaly}})$
4: $\quad (M_{t+1}, K_{t+1}) \leftarrow F(\hat{M}, \hat{K})$ {self-correct}
5: **end while**

---

Supporting the two functions is a shared knowledge accretion function that detects anomalies, generates abstractions and binds fresh knowledge into memory.

---

**Algorithm 3** ACCRETE $(X^*, M, K, \alpha_{\text{anomaly}})$

---

**Require:** $X^*$: a representation, $M$: memory, $K$: knowledge, $\alpha_{\text{anomaly}}$: anomaly threshold
**Ensure:** $M, K$
1: **if** $C(M, X^*) > \alpha_{\text{anomaly}}$ **then**
2: $\quad K' \leftarrow \pi(M, X^*)$ {generate new high-order structures}
3: $\quad$ **if** $K' \neq \emptyset$ **then**
4: $\quad\quad K \leftarrow K \cup K'$ {self-organize and evolve to assimilate new structures}
5: $\quad$ **end if**
6: **end if**
7: **return** $(M, K)$

---

## B. Comparison with the Kahneman Model

The interpretation of the dual system model of SOKA has nominal deviations from the Kahneman model (Kahneman, 2011). Whereas the following table outlines the features of the two system in this Kahneman model:

| System 1 | System 2 |
|---|---|
| Does not require working memory | Requires working memory |
| Autonomous | Cognitive decoupling; mental simulation |
| Fast | Slow |
| High capacity | Capacity limited |
| Parallel | Serial |
| Nonconscious | Conscious |
| Biased responses | Normative responses |
| Contextualised | Abstract |
| Automatic | Controlled |
| Associative | Rule-based |
| Experience-based decision making | Consequential decision making |
| Independent of cognitive ability | Correlated with cognitive ability |

The SOKA model, achieves the same system objectives by redistributing features along functional lines.

| Reactive, $\kappa$ | Deliberative, $\Delta$ |
|---|---|
| Shared memory + episodic cache | Shared memory + incubation space |
| Fast (response time constrained) | Slow (time-slice limited) |
| Serial+Workers | Parallel |
| Conscious | Non-conscious, memory barrier |
| Interactive | Normative process, non-interactive |
| Contextualised | Abstract |
| Autonomic | Stochastic |
| Associative | Rule-based |
| Experience-based decision making | Consequential decision making |

## C. Case Study: Mary Shelly's Frankenstein

In the classic Gothic/science fiction novel *Frankenstein; or, The Modern Prometheus* (1818) by Mary Shelley, a scientist creates a sentient, hideous creature from body parts, leading to tragedy for both creator and creation. This story, of a lonely monster seeking vengeance, explores themes of ambition, responsibility, creation, and prejudice. When visualized as limited metagraph of *linguae primae* constructs of (a) attributions and (b) generalizations, the themes and narrative structures of the main characters emerge from the noise as intricate floret-like structures of knowledge, demonstrating both role interconnections and the depth of character

development in the novel.

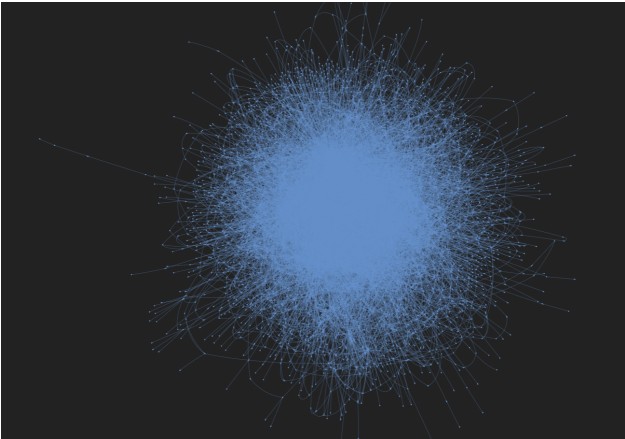

*Figure 8.* The narrative structure of Frankenstein, the novel, extracted as a dense metagraph of information from which it becomes possible to sift and distill both factual and abstract knowledge.

Our methodology: (i) We theorized that language structures exist in all rich texts. (ii) Exploring this hypothesis, we generated a graph that encoded the classic Frankenstein, extracting only statements of attribution and generalization. This transformed the text into a dense metagraph of concept that emerge as interlinked as conception as shown in (Figure 8). (ii) Reducing these complex structures to meaningful concepts required graph analysis. We unraveled the recursive 'bowtie' structures, commonly found in large graphs, to uncover hidden patterns. For this, we employed graph analysis to extract *Strongly Connected Component* (SCC) (Chakrabarti &Faloutsos, 2012). This revealed complex narrative structures within the text (Figure 9). (iii) Studying each character in the novel carefully and comparing their conception in the metagraph validated our hypothesis that key features of the principal characters are revealed by their conceptions: *Elizabeth* - Figure 10, *Mother* - Figure 11, *Father* - 12.

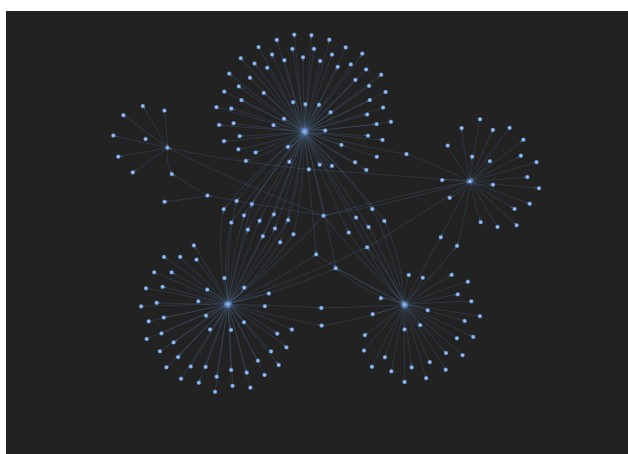

*Figure 9.* The narrative structure filtered on the main characters became more pronounced, showing their individual description as well as their inter-relationships in the novel.

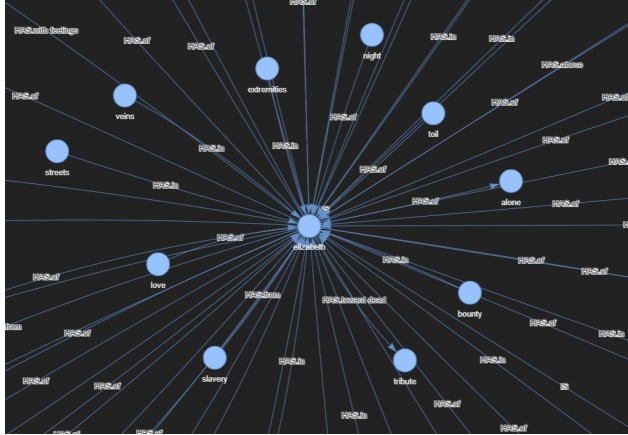

*Figure 10.* The conception of Elizabeth, a principal character in the novel.

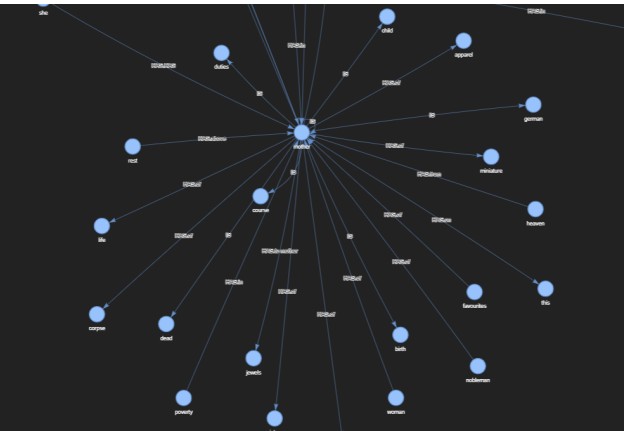

*Figure 11.* The conception of Victor Frankenstein's mother, a minor character in the novel.

Although our findings are as yet inconclusive, owing to the challenge of isolating the considerable noise in the data, this early research provides a glimpse of the possibilities. No facility for anomaly detection or consistency checking was employed. Nevertheless, the structures that emerged from this minimal processing indicated the power of our *linguae primae* to codify and capture the knowledge hidden in plain text into symbolic constructs with geometric features pliable using topological calculus. This research also showed how the essential components of SOKA: input operator $I$, memory structure $M$, representation operator $\mathcal{R}$, language primitives $\mathcal{L}$, and update operator $U$, can come together to perceive and encode streams on input.

## D. An Example of Analytical Reasoning

Concept invention and schema emergence require advanced analytical reasoning to synthesize abstract conception. This is the role of $G$, the Knowledge Structure Synthesis Operator in Equation.9. A simple example of how this could work is outlined here. Note that the entire process of reasoning can

*Figure 12.* The conception of Victor Frankenstein's father, a central character in the novel, shows the intricate details of character development.

be conducted using graph traversal and set operations over subgraphs, if the concepts are represented as a metagraph. For simplicity, we annotate the graph as proposition groups associated with an entity.

### D.1. Prior Rules & Facts

New knowlege emerges from hypothesis; hypothesis formed from the inspection of existing facts and the application of logical rules to it. To explain how, consider the following prior facts and rules known to the system.

#### D.1.1. PRIOR FACTS

Prior facts are an extracted slice of the knowledge in the memory. We identified $G$, the Knowledge Structure Synthesis Operator in Equation.9 as responsible for its synthesis. Note that each atomic statement is a *linguae primae* attribution or classification.

```
scott {
 is man
 is creature
}

frankenstein {
 is man
 is creature
 has lab
}
```

#### D.1.2. PRIOR RULES

Prior rules must themselves be knowledge accessible to the process of deliberation. These are hence also atomic

statement in *linguae primae* represented as operation. Most are presumably *a genetia*, originally preprogrammed into the system.

```
rule1(x, y, z): {
    if x is y and x is z
    -> y is z
}

rule2(x, y, z): {
    if x is y and x has z
    -> y has z
}
```

### D.2. Reasoning

Reasoning involves inspecting the locality of a concept (*man*), in other words its conception, to arrive at propositions.

*By Induction*: man is a creature, because all prior facts satisfy the proposition.

$$\forall x \left( Man(x) \rightarrow Creature(x) \right) \tag{10}$$

*By Abduction*: man may have a lab, because atleast one man (frrnkenstein) has a lab.

$$\exists x \left( Man(x) \wedge HasLab(x) \right) \tag{11}$$

The two synthetic propositions, we arrive at the following new abstract relationships in the metagraph:

```
man {
 is creature
 ~has lab
}
```

Consequently if the following proposition arrives as input to the system:

```
edward{ is man }
```

It may be *deducted* conditionally that:

```
edward{
 is man
 is creature
 has lab
}
```

## Glossary

**autopoiesis** Greek αὐτο (auto) and ποίησις (poiesis): 'self' and 'creation, production'. An autopoietic system can maintain itself by creating its own parts, as exemplified by a biological cell. https://en.wikipedia.org/wiki/Autopoiesis . 4

**concept** An abstract, intensional unit of meaning that supports generalization across instances. Formally, you can treat a concept as a latent predicate/classifier or schema (C) that induces (i) an extension (the set/distribution of things it applies to) and (ii) constraints on how it composes with other concepts. https://en.wikipedia.org/wiki/Philosophyd_of_mind . 7, 13

**conception** A particular realization/interpretation of a concept within an agent or theory—i.e., the concrete representational and inferential commitments that make the concept usable. Where a *concept* is "what is meant," a *conception* is "how it is modeled/implemented" (features, rules, exemplars, causal model, embeddings, etc.).. 5, 13, 14

**deliberative reasoning** a slower, reflective process (Δ) that operates over the same metagraph memory to construct, test, reorganize, and legitimize higher-order concepts and schemas through abstraction, anomaly resolution, and empirical validation, enabling the long-term evolution of living knowledge . 6–8, 12

**language primitives** The smallest typed atoms from which language structures are built. Depending on the level of analysis, primitives may be tokens/subwords, lexical items, variables/constants, function symbols, relation symbols, or core constructors (e.g., application/abstraction in lambda calculus). The key property is that primitives have stable roles in composition (type/arity/selectional constraints), not just surface identity.. 5, 6

**language structures** The organized forms that constrain how primitives and operators can combine—syntax and higher-order organization. Concretely: strings with grammar, parse trees, dependency graphs, lambda terms, discourse graphs, programs/ASTs. They define well-formedness, conceptual relationships and enable systematic compositionality.. 4, 5

**production rule** a rewrite rule that replaces some symbols with other symbols, a finite set of which is the main component in the specification of a formal generative grammar. . 4

**reactive reasoning** a fast, sensory-facing process (κ) that continuously assimilates observations into the shared knowledge substrate and drives an agent's immediate action using accumulated experience, operating as a real-time reflex loop grounded in the current state of living knowledge. 6–8, 12

