# OpenReview forum: "Position: Living Knowledge, Foundations for General Intelligence"
_ICML.cc/2026/Position_Paper_Track — Submitted to ICML 2026 Position Paper Track_

### Official Review · Reviewer_rhdf · 2026-03-09

**Significance:** 3
**Argument Clarity:** 2
**Rating:** 3
**Confidence:** 3

**Questions:**

1. The paper proposes seven language atomics as the foundation of knowledge representation. Can the authors provide theoretical or empirical evidence to justify that these atomics constitute a minimal and sufficient set?

2. Compared with existing representational structures, such as property graphs, RDF graphs, and hypergraphs, what are the practical advantages of the metagraph? Is there any formal analysis or complexity-based justification?

3. How does SOKA acquire new primitives or relations? Does it rely on manually designed linguistic structures?

**Alternative Views Section:**

Yes

**Compliance With Llm Reviewing Policy A Conservative:**

Affirmed.

**Discussion Potential:**

3

**Final Justification:**

Thank you for the detailed and thoughtful rebuttal. I appreciate the authors’ effort in clarifying aspects of the work and further elaborating on the motivation and contributions. However, after carefully reviewing the rebuttal, my overall assessment remains unchanged. While the paper has certain merits, the key concerns raised in my original review have not been sufficiently addressed. Therefore, I still consider the paper to be on the borderline of rejection, and I will maintain my original score.

**Paper Summary:**

The paper presents a position on the foundations of knowledge representation for Artificial General Intelligence (AGI), arguing that achieving general intelligence requires a “living knowledge” system, namely a knowledge structure capable of self organization, continuous evolution, and the generation of new concepts.

The authors argue that current mainstream AI approaches, including deep learning, neural networks, knowledge graphs, and neuro symbolic systems, face fundamental limitations in knowledge management. These limitations include insufficient long term memory, limited capacity for continual learning, and knowledge structures that rely on predefined schemas. Such constraints restrict the ability of AI systems to accumulate knowledge and support concept evolution in open environments.

To address these limitations, the paper proposes a language centered knowledge representation framework. Within this framework, knowledge is assumed to be decomposable into three basic types of linguistic structures, which are further abstracted into seven language atomics: attribution, classification, relation, collection, denotation, operation, and transition. These atomics are argued to constitute the minimal compositional basis for universal knowledge representation.

**Position:**

Yes

**Position In Title:**

No

**Related Work:**

3

**Strengths And Weaknesses:**

**Pros:**

1. The paper focuses on knowledge representation for general intelligence, which is a long standing and important research problem.

2. The central position of the paper is relatively clear.

3. Although the paper is primarily conceptual in nature, it still proposes several concrete structural elements.

**Cons:**

1. Position is not sharply articulated. The causal relationships and the necessity among the key propositions are insufficiently justified.

2. The paper provides almost no empirical support. The only content resembling an experiment is the metagraph visualization of the Frankenstein novel text. However, this is merely a visualization example rather than a substantive empirical evaluation.

3. Some of the concepts proposed in the paper resemble ideas that already exist in prior literature. The manuscript does not clearly articulate the genuine novelty of the work in comparison with existing studies.

**Support:**

2

---

> ### Author Rebuttal · Authors · 2026-03-30
>
> Please refer note to all reviewers in response to y4uh
>
> ## Rebuttal
>
> Q1:	The paper proposes seven language atomics as the foundation of knowledge representation. Can the authors provide theoretical or empirical evidence to justify that these atomics constitute a minimal and sufficient set?
>
> [rhdf:Q1] Acknowledged. No evidence of minimality nor sufficiency is provided in the paper. You are also right – the 7 is incomplete; but they cover a usefully large area of the descriptive space. We believe such a proof may not exist formally; would not a semantic bound if proven by a formal system violate Godelien incompleteness. The proof may have to be empirical. Refer to [yWfs:Q2] for the particular reasons for our choice.
>
>
> Q2:	Compared with existing representational structures, such as property graphs, RDF graphs, and hypergraphs, what are the practical advantages of the metagraph? Is there any formal analysis or complexity-based justification?
>
> [rhdf:Q2] Yes, you are correct. Property graphs, metagraphs and hypergraphs certainly allow you to represent the data, just as well as graphs may be represented in relational forms themselves. In all - graphs, metagraphs and hypergraphs, we saw a problem in these forms the set of vertices and the set of edges are disjoint sets. That meant that graph traversals between the two domains required a ‘switch’; properties fill this gap. But this complicated the query, because prior knowledge of the domain switch is often necessary. Metagraph is suggested as an alternative (opening it to debate), where all edges are first class vertices.
>
> For example - scott.father.description can be efficiently represented in space as: (a) a description of scott’s father Edward (b) a description of scott as a father (c) a description of fatherhood. As such, father.description can still potentially capture what a ‘male parent’ means  or as an abstract concept like the ‘father of a nation’. However, traversal along the graph for concepts for discovery (deliberation process) does not require drastic specialization of search heuristics.
>
> Q3.How does SOKA acquire new primitives or relations? Does it rely on manually designed linguistic structures?
>
> [rhdf:Q3] We don’t have a complete answer. But we can attempt to share what we know.
> Let’s first reframe this question as follows: How can a given metagraph be transmuted into something larger than itself?  There are atleast two ways.
>
> 1. Induction and abduction (CS. Pierce) as shown in Appendix D.
> 1. Abstractions from dualities
>
> The latter needs further explanation. One property we observe about the several primitives is that a primitive proposition often exist as a duality.
> eg: a set of classification existing as ground truth
> ```
> abel {
>    is man
> }
>
> kane{
>    is man
> }
> ```
> also implies a collection:
> ```
> man {
>   contains [abel, kane]
> }
> ```
>
> The collection can be conditionally inferred from the classification by abduction. This is what the slice and fit process is expected to do in SOKA – synthesize abstractions and test their cogency, before empirically legitimizing it. Thus schemas can emerge through iterative abstraction of known truth and abstractions themselves.
>
> ## Final Note
> Given its current form, the paper has an architectural and philosophical position, perhaps better suited for a philosophy-oriented journal that already covers topics at the nexus of AI/technology/foundations work. However, we believe broader discourse on the topic of living knowledge is desirable. As such, if the paper can be accepted with minor revisions, here's what would be our course of action:
> * Narrow and soften the central thesis by reframing that scope of the paper and limit it to the language constructs
> * Reframe the contribution section explicitly to situate it as a position-paper contribution
> * Clarify that SOKA is a concrete research hypothesis and agenda, only arguing that open-ended intelligence like SOKA is required for dynamically self-revising knowledge representations.
> * Minor annotation with cross-references to examples the Appendix to clarify abstract concepts with examples.
>
> Your feedback is well received. Thank you.

---

> > ### Author Rebuttal · Reviewer_rhdf · 2026-04-02
> >
> > Thank you for the detailed and thoughtful rebuttal. I appreciate the authors’ effort in clarifying aspects of the work and further elaborating on the motivation and contributions. However, after carefully reviewing the rebuttal, my overall assessment remains unchanged. While the paper has certain merits, the key concerns raised in my original review have not been sufficiently addressed. Therefore, I still consider the paper to be on the borderline of rejection, and I will maintain my original score.

---

### Official Review · Reviewer_yWfs · 2026-03-10

**Significance:** 3
**Argument Clarity:** 3
**Rating:** 3
**Confidence:** 3

**Questions:**

Please see the weakness

**Alternative Views Section:**

Yes

**Compliance With Llm Reviewing Policy A Conservative:**

Affirmed.

**Discussion Potential:**

2

**Final Justification:**

I appreciate the authors’ detailed rebuttal, which demonstrates their deep reflection on the issues raised in this paper and has gradually helped me understand its intended significance. While the current version may not yet present the explanations as clearly as possible, I believe the questions the authors propose remain valuable. Therefore, I am inclined to raise the score to 3.

**Paper Summary:**

This paper argues the necessity of “Live Knowledge” to build AGI, which is a dynamic and easily updatable knowledge representation method that can effectively empower current AI systems. This paper draws insights on the philosophical paradigms of Aristotle and Descartes regarding cognition, and introduces the SOKA framework which is built upon seven linguistic primitives as the atomic units of knowledge.

**Position:**

Yes

**Position In Title:**

Yes

**Related Work:**

3

**Strengths And Weaknesses:**

Strengths:

1.	The idea of introducing dynamically evolving knowledge to replace current rigid knowledge representation to empower AI system is reasonable and convincing.

2.	The “Alternative View” section provides an excellent survey of existing works and thus is comprehensive.

3.	The paper is well structured and clearly written.

Weaknesses:

1.	As this paper stated, one of the basis of proposed SOKA framework, seven atomic symbols representation, is not rigorously proven but only relying on a qualitative case study. I think this severely reduces the feasibility of the framework proposed in the paper.

2.	The technology section lists a series of open questions, including the minimal schema set, the system's initial state, etc. These questions are central to whether SOKA can move from blueprint to practice. Leaving them all as open questions essentially acknowledges that the architecture is nearly impossible to implement with current technology, making the paper read more like a "challenge" than a "roadmap."

**Support:**

2

---

> ### Author Rebuttal · Authors · 2026-03-30
>
> Please refer note to all reviewers in response to y4uh
>
> ## Rebuttal
>
> [yWfs:Q2] Acknowledged. The seven symbols are only proposed as a *candidate* minimal basis and minimality and sufficiency remain empirical and theoretical questions. We should have been explicit about its candidature rather than the current tone that they are definite.
>
> Schema minimality/sufficiency:
> At several forums where we present this work, the audience raised the question that perhaps only relations are ever required (a position rooted in ideas that gave us Prof. Peter Chen’s Entity-Relationship diagrams). The certainly could be richer on one side or as minimal as a ER model on the other extreme. Here we cede that from a linguistic perspective, modals, counterfactuals, tenses, generics, existentials are a few types of statements that do not easily fit the schema. Once again, the point is to engage a necessary and critical discourse.
>
> SOKA Feasibility:
> Acknowledged. But we think generative technology (LLM, VML) may be closer to realizing a useful architecture that we give it credit for.
>
> --------------------
> Example test case: Consider the following test where we attached the research paper to the chat post presented to CoPilot and asked it to analyze a simple statement.
>
> Priming Query: Study this paper and describe lingua primea for me
>
> ```
> OUTPUT: …. Some priming output used to assure that the LLM understands the paper. We find that language models with lower 7-billion parameters do not.
> ```
>
> Test Query: Extract all the propositions in 7 primitives of this sentence: Mary cut the bread with a knife.
>
> ```
> OUTPUT: ……
>
> Entities:
> * m = Mary
> * b = bread
> * k = knife
> * e = cutting event
>
> Attribution:
> * b is cut              (post-state)
>
> Classification:
> * b is bread
> * k is knife
> * m is Mary   (named individual / optional as type anchor)
>
> Relation:
> * m is-agent-of e
> * b is-patient-of e
> * k is-instrument-of e
>
> Collection:
> * none explicit
>
> Denotation:
> * none explicit
>
> Operation:
> * cut(m, b, k)
> * bread -> cut-bread   (transformation view)
>
> Transition:
> * e causes b to become cut
> ```
>
> Based on experiments like these, we think that LLMs are (a) capable of generating both lingua primae atomics and (b) fills the critical gap of *G* the Knowledge Structure Synthesis Operator described in Appendix A. The remainder of the heuristic is made possible by topological set operators with semantic monads on a graph.
>
> As a side note, it is worth mentioning that a metagraph representation does not necessarily replace a language technology like a LLM. But it can certainly be useful as a rich, human-qualified, repeatable and explainable semantic bound of an LLM output, in very much the way a RAG system serves that role. This is how we see living knowledge complementing the conventional frontier AI systems that serve us today.

---

> > ### Author Rebuttal · Reviewer_yWfs · 2026-04-02
> >
> > I appreciate the authors’ rebuttal and the illustrative example demonstrating that LLMs can parse and generate atomic knowledge units. However, this raises a fundamental question regarding the necessity of the SOKA/living knowledge framework. If modern LLMs are already capable of identifying, analyzing, and generating these primitives, it is unclear what additional functional value SOKA provides beyond the LLM itself.

---

> > > ### Author Response · Authors · 2026-04-06
> > >
> > > It is unfortunate that SOKA became the central focus of the paper, when it was never the intent; lingua primae was. That said, I think I accept the panel's final decision and respectfully rest my rebuttal. I won't pursue this further.
> > >
> > > However let me can respond to your question. It is a meaningful one.
> > >
> > > Recent agentic trends - posts from Andrej Karpathy (https://x.com/karpathy/status/2039805659525644595) and software like OpenClaw, suggest that the value of living knowledge is as a complement to LLM. In each of these cases, a markdown file was uses and a repository for knowledge that is iteratively refined.
> > >
> > > SOKA's function is two fold: a) a suggestion that a dual mode architecture can achieve schema labile model representation. b) Unlike the agentic equivalents, it generates a discrete mathematical construct, a graph - reproduceable, explainable, bounded and amenable to set operator semantics. This we felt would lead to explainable and safer outcomes.
> > >
> > > Here, perhaps I should emphasize the origins of our work. Several of my co-authors are involved in the machine interpretation of maritime regulations (COLREGs being just one of many that interoperate/interplay in maritime transport and traffic).  One immediate problem in our domains is that rules are often open-ended and left open to the judgement of the ship's captain. How a rule i to be interpreted is circumstantial - depending on the situation, location, business priorities, risk tolerance etc.  Interpreting them effectively requires years of experience under an experienced crew. How do you teach that to a machine? How do you test its bounds? How do you bound what it can do if it is uncertain? How will it know if it has random walked into a 'hallucination'?
> > >
> > > Increasingly, neural networks are being used to control robotics. Yet we have know way to draw bounds around it. Our current theory is that we may have to tap into knowledge hidden in documents - legal claims, insurance contracts, witness reports, media reports, crew logs. Simply relying on video or sensor logs, to train the machines leaves too much to uncertainty.
> > >
> > > SOKA emerged out of a need to capture this latent knowledge and experience into a discrete, verifiable mathematical that is more easily evaluated using conventional software engineering practices. This led us to ask the question: are there cognate concepts... conceptual eigens, characteristic roots that we collectively share as experience? if so, how can they be captured using a machine algorithm?
> > >
> > > The paper attempts to invite readers into a discussion about these questions.
> > >
> > > I have no conclusive answer, but I hope it partially answers atleast the question of situation: the place of SOKA vs. LLM.

---

### Official Review · Reviewer_e65a · 2026-03-11

**Significance:** 2
**Argument Clarity:** 3
**Rating:** 2
**Confidence:** 3

**Questions:**

I don't have questions for the authors.

**Alternative Views Section:**

Yes

**Compliance With Llm Reviewing Policy A Conservative:**

Affirmed.

**Discussion Potential:**

4

**Final Justification:**

I appreciate the authors for criticism about the potentiality of discussion this paper has, however the lack of concreteness for moving to SOKA-like methods and potential ambiguous pieces makes me keep my original score.

**Paper Summary:**

This paper expands on the idea that to unlock AGI, a shift to the so-called living knowledge should be done to allow for artificial systems to mimic biological intelligence. The thesis mainly revolves on philosophical arguments, contrasting existing ML paradigms, and gets to the idea of introducing a self-organizing knowledge architecture (SOKA). This should be created using metagraphs and combining both neural and reasoning processes.

**Position:**

Yes

**Position In Title:**

Yes

**Related Work:**

1

**Strengths And Weaknesses:**

### STRENGTHS

The idea is entirely novel to me, and looking to new candidate architectures for improving in "AGI benchmarks" (see Hendrycks et al. 2025) should be welcomed and discussed in the community. The recollection of both philosophical, cognitive, and neurolinguistic perspectives of the past is a good contribution, which can allow reconsidering fundamental pieces of modern AI and think to alternatives more philosophically grounded (in contrast to the leading "tabula rasa" approach of current LLMs).

### WEAKNESSES

The main weakness of this paper is that it is not concrete. Comparing to other position papers, the transitioning to a different architecture (not entirely neural or neuro-symbolic) is not extensively justified. While the paper revolves on presenting the motivation for SOKA, the position mainly informs of the authors' perspective but do not capture a wide audience and does not connect to works in the field.
While I appreciate that it is worth discussing what philosophical commitment AI practitioners should adhere to (and this is clearly raised by the position), there is no evidence supporting the transition to SOKA-like models. This weakens the position, as the abstract nature of the proposal does not imply something already actionable and grounded on evidence.

No call for action is presented. No prototypes are included for SOKA, which makes me think that a position of this sort is premature.

### POTENTIAL DIRECTIONS FOR IMPROVEMENT

Given that the paper thesis rises the debate on which model of AI we should adhere for meeting AGI expectations, it would be interesting to observe even prototypical implementations of SOKA and compare it to existing models. That would be a great improvement for the position, helping substantiate many claims about robustness, reasoning, and continual learning. These three aspects are well-studied in the literature, but no explicit connection is drawn to them. If SOKA is a key for AGI, it would be ideal to observe improvement in tasks where other methods fail.

**Support:**

1

---

> ### Author Rebuttal · Authors · 2026-03-30
>
> Please refer note to all reviewers in response to y4uh
>
> ## Rebuttal
>
> [e65a:Q- WEAKNESSES]
> On Transition to SOKA-like architecture, refer to answer to [y4uh:Q4]
> On call for action: Yes. We should have been explicit. The SOKA architecture and the 7 primitives were presented as starting points for how Living Knowledge may be realized. Appendix A. Formal Sketch of a Knowledge Architecture, was our first draft of how the program would work. Also note that our response to [yWfs:Q2] refers to a plausible use of LLMs to generate cogent hypothesis about the external world in primitive form. This is the only obscure part of the heuristic of the system sketched in the appendix.
> Note that SOKA is not suggested as the only way forward. We think there may be better approaches to living knowledge.
>
> [e65a:Q- IMPROVEMENTS] Agreed. An implementation of SOKA-like architecture is a desirable. In particular they test the falsifiability of a few of our hypothesis.
> * If a fixed-schema architecture can match SOKA-style systems on continual concept revision tasks without growing representational machinery, that weakens our thesis.
> * If dynamic metagraph restructuring improves long-horizon adaptation and conflict resolution over static memory baselines, that supports the thesis.
> * If the seven atomics fail to cover common revision scenarios without extensions.

---

> > ### Author Rebuttal · Reviewer_e65a · 2026-04-03
> >
> > I thank the authors for the response. However, at the current stage, the framework is not very knowledgeable of how SOKA  should work and why its building blocks are fundamentally needed. I believe this idea can be further investigated before presenting it as a position for "transitioning" ml models backbones. Finally, I will keep my score.

---

### Official Review · Reviewer_y4uh · 2026-03-18

**Significance:** 3
**Argument Clarity:** 2
**Rating:** 3
**Confidence:** 3

**Questions:**

Please refer to weakness.

**Alternative Views Section:**

Yes

**Compliance With Llm Reviewing Policy A Conservative:**

Affirmed.

**Discussion Potential:**

3

**Paper Summary:**

This paper proposes that general artificial intelligence requires living knowledge and advocates that knowledge systems should possess the ability to self-organize, grow and reorganize. To this end, this paper takes language as the knowledge primitive and combines the meta-graph structure to design a self-organizing knowledge architecture (SOKA) with dual reasoning loops of reactive and deliberative reasoning.

**Position:**

Yes

**Position In Title:**

Yes

**Related Work:**

3

**Strengths And Weaknesses:**

Strength:

1. The paper identifies a key limitation of current large language models: the static nature of encoded knowledge. It argues that achieving general AI requires living knowledge, which is of great significance for the continuous update and knowledge evolution in LLMs.

2. The authors propose a novel SOKA framework for dynamically organizing and updating knowledge, which provides an approach for realizing living knowledge.

Weakness:

1. The paper argues that general intelligence requires living knowledge, but the necessity for this claim is insufficiently substantiated. While static knowledge representations are mentioned as a limitation, this paper does not provide concrete examples or systematic demonstrations showing how traditional AI systems fail in such tasks requiring continual learning and long-term knowledge retention/update. Incorporating illustrative examples  or empirical observations would enhance the persuasiveness of the argument.

2. Core concepts, including language atoms, metagraphs and living knowledge, are defined with a high degree of abstraction and are difficult to understand intuitively. Providing a concrete example that systematically demonstrates how these elements are constructed and interrelated would help readers understand the framework.

3. To my understanding, the deliberative reasoning component of the SOKA architecture is central to living knowledge, supporting knowledge self-organization, growth, and creation. However, the current description is too abstract and fails to clarify how abstract knowledge is generated, empirically validated, or reintegrated into memory. This reduces clarity and undermines the plausibility of the proposed architecture.

4. Beyond being difficult to fully understand, the SOKA architecture lacks discussion or evidence regarding its practical feasibility. While the concept of living knowledge is theoretically compelling, it remains unclear whether the proposed SOKA implementation can be realized effectively in practice. Without validation or feasibility analysis, it is difficult to assess whether this architecture would meaningfully advance research in general intelligence.

**Support:**

2

---

> ### Author Rebuttal · Authors · 2026-03-30
>
> ## Note to all reviewers
>
> Your critique is much appreciated. We are glad that your response was the kind of strong discourse we expected and wanted to kindle.
>
> As a preface to our rebuttal, we want to clarify one subject: Our central theme for the paper was knowledge representation, in particular the thesis around the existence of lingua prima, a primordial language in all knowledge systems. Early on however, it became clear to us that a position statement arguing for a new language theory cannot be situated in isolation; it had to be backed by at least supporting theories on application (architecture) and representative data structures (metagraph) for discourse to advance. As such, our intent with this position paper is to encourage discourse in the original spirit of your stated goal - "papers that stimulate constructive, civil discussion on timely topics that need our community’s attention.  Controversial topics are welcome". We chose to present the paper in this spirit. This paper is by not stretch of the imagination a full paper.
>
> Hence, we acknowledge that sections of the paper including SOKA may have overreached and could be toned down in commitment. They are only sketched out as a possible research directions for a constructive discourse around living knowledge. In the interest of economy, given the 9 page limit, we also have delivered a paper that is abstract and at places insufficiently concrete.
>
> In the following response, we have used a scheme referring to your reviewer code and the specific question number to distinguish your queries. Some of them may cross reference each other on overlapping questions.
>
> ## Rebuttal
> Q1: The paper argues that general intelligence .... the persuasiveness of the argument.
>
> [y4uh:Q1] - Yes. We can do better. Revised, we would say that: Whereas Traditional AI can demonstrate a capacity for countinual learnind and long-horizon knowledge assimilation, it is deficient and/or delinquent in three areas –  (a) semantic closure, a system or language in which it can represent its own meanings and rules within itself. (b) semantic bounds – a system that understands the boundary of what it can do and not (c) semantic introspection – a system that can interpret its own existence and rules from within itself.
>
> Q2: Core concepts, including language atoms, ...framework.
>
> [y4uh:Q2] – Yes. The case study Appendix C and the example in Appendix D, needs to be connected with a few more annotations. The example in D might have been represented as a visual symbols in line with lingua primae, demonstrating prior rules, and reasoning are homologous to geometric operations on primitive symbols.
>
> Q3:	To my understanding, ... the proposed architecture.
>
> [y4uh:Q3] – Acknowledged. But we had more modest goals that to deliver a new cognitive architecture.  The main goals of this paper is to swing the discourse back to refocus on symbolic representations of knowledge as a necessary complement to contemporary neural networks architectures, not as a replacement for them.
> * A position arguing that open-ended, continual, concept-evolving intelligence is unlikely to be achieved with purely static knowledge representations alone.”
> * Proposing living knowledge as a testable design hypothesis for future AGI-oriented systems.
> * SOKA only presented as a call-to-action research program and suggested only as an architectural sketch, not a complete validated system. It is an invitation to encourage precisely the debate each of the reviewers have raised here.
> To that end, ‘Section 7 - Technology for Living Knowledge’ in its entirety was conceived as lines of inquiry for future research, suggesting that both datastructures and architecture will need to be revisited to realize autopoesis.
>
> Q4:	Beyond being difficult ... in general intelligence
>
> [y4uh:Q4] - Yes. We are of a similar opinion. Validation and feasibility analysis is required and SOKA if appropriate, requires an entire paper for its treatment thoroughly. But for a position paper we wanted to emphasize that  (a) a schema labile architecture is possible (b) however, a dual process architecture is a necessary consequence. (c) that tautologies cannot exist in such a system that allows for the existence of contradictions (of proposition) (d) truths emerge from cognitive dialectics in the system, but they can only be legitimized by natural selection (from experience) necessitating both latent, emergent and contradictable truth.
>
> Your feedback is well received. Thank you.

---

> > ### Author Rebuttal · Reviewer_y4uh · 2026-04-05
> >
> > Thank you for your response. However, I remain concerned about the focus of the paper and the feasibility of the SOKA architecture. Given the title, “Living Knowledge, Foundations for General Intelligence”, a position paper should primarily clarify the definition, advantages, necessity, and potential impact of living knowledge, yet these aspects are underdeveloped. Moreover, the presentation of SOKA is difficult to follow, and its feasibility is hard to assess; even as a conceptual architectural sketch, it is challenging to evaluate or endorse. Therefore, I will keep my original score.

---

### Decision · Program_Chairs · 2026-04-30

**Decision:**

Reject

**Comment:**

The review process highlighted a tension between the paper's philosophical ambition and its technical concreteness.
•	Substantiation of "Necessity": the claim that AGI requires living knowledge was insufficiently substantiated. The authors clarified by identifying three specific areas where traditional AI is deficient: semantic closure (representing its own rules), semantic bounds (understanding its own limits), and semantic introspection (interpreting its own existence)
•	Abstractness vs. Clarity: A major critique was the "high degree of abstraction" in defining core concepts like language atoms and metagraphs, making them difficult to understand intuitively. In particular, the description of the deliberative reasoning loop ($\Delta$) was too vague regarding how knowledge is actually generated and validated.
•	Practical Feasibility: There was significant concern regarding the lack of evidence that the SOKA architecture could be effectively implemented in practice.
Although the authors argued that the paper is intended to stimulate high-level discourse on knowledge representation rather than provide a full technical specification, overall the paper has not provided sufficient novelty with more technical feasibility analysis.